# Interpretation of pre-morbid cardiac 3T MRI findings in overweight and hypertensive young adults

Gert J. H. Snel [1]*, Riemer H. J. A. Slart[2,3], Birgitta K. Velthuis[4], Maaike van den Boomen[1,5,6], Christopher T. Nguyen[5,6], David E. Sosnovik[5,6,7], Vincent M. van Deursen[8], Rudi A. J. O. Dierckx[2], Ronald J. H. Borra[1,2], Niek H. J. Prakken[1]

1 Department of Radiology, Medical Imaging Center, University Medical Center Groningen, University of Groningen, Groningen, The Netherlands, 2 Department of Nuclear Medicine and Molecular Imaging, Medical Imaging Center, University Medical Center Groningen, University of Groningen, Groningen, The Netherlands, 3 Faculty of Science and Technology, Department of Biomedical Photonic Imaging, University of Twente, Enschede, The Netherlands, 4 Department of Radiology, University Medical Center Utrecht, University of Utrecht, Utrecht, The Netherlands, 5 Department of Radiology, Athinoula A. Martinos Center for Biomedical Imaging, Massachusetts General Hospital, Harvard Medical School, Boston, MA, United States of America, 6 Cardiovascular Research Center, Massachusetts General Hospital, Harvard Medical School, Boston, MA, United States of America, 7 Division of Health Sciences and Technology, Harvard-MIT, Boston, MA, United States of America, 8 Department of Cardiology, University Medical Center Groningen, University of Groningen, Groningen, The Netherlands

* g.j.h.snel@umcg.nl

**Data Availability Statement:** All relevant data are within the paper and its Supporting Information files.

## Abstract

In young adults, overweight and hypertension possibly already trigger cardiac remodeling as seen in mature adults, potentially overlapping non-ischemic cardiomyopathy findings. To this end, in young overweight and hypertensive adults, we aimed to investigate changes in left ventricular mass (LVM) and cardiac volumes, and the impact of different body scales for indexation. We also aimed to explore the presence of myocardial fibrosis, fat and edema, and changes in cellular mass with extracellular volume (ECV), $T_1$ and $T_2$ tissue characteristics. We prospectively recruited 126 asymptomatic subjects (51% male) aged 27–41 years for 3T cardiac magnetic resonance imaging: 40 controls, 40 overweight, 17 hypertensive and 29 hypertensive overweight. Myocyte mass was calculated as (100%–ECV) * height$^{2.7}$-indexed LVM. Absolute LVM was significantly increased in overweight, hypertensive and hypertensive overweight groups (104 ± 23, 109 ± 27, 112 ± 26 g) versus controls (87 ± 21 g), with similar volumes. Body surface area (BSA) indexation resulted in LVM normalization in overweights (48 ± 8 g/m$^2$) versus controls (47 ± 9 g/m$^2$), but not in hypertensives (55 ± 9 g/m$^2$) and hypertensive overweights (52 ± 9 g/m$^2$). BSA-indexation overly decreased volumes in overweight versus normal-weight (LV end-diastolic volume; 80 ± 14 versus 92 ± 13 ml/m$^2$), where height$^{2.7}$-indexation did not. All risk groups had lower ECV (23 ± 2%, 23 ± 2%, 23 ± 3%) than controls (25 ± 2%) (P = 0.006, P = 0.113, P = 0.039), indicating increased myocyte mass (16.9 ± 2.7, 16.5 ± 2.3, 18.1 ± 3.5 versus 14.0 ± 2.9 g/m$^{2.7}$). Native $T_1$ values were similar. Lower $T_2$ values in the hypertensive overweight group related to heart rate. In conclusion, BSA-indexation masks

**Funding:** G.S. and N.P. were supported by a grant from the Dutch Heart Association (2016T042, https://www.hartstichting.nl/). The funders had no role in study design, data collection and analysis, decision to publish, or preparation of the manuscript.

**Competing interests:** The authors have declared that no competing interests exist.

hypertrophy and causes volume overcorrection in overweight subjects compared to controls, height$^{2.7}$-indexation therefore seems advisable.

## Introduction

Western lifestyle is increasingly characterized by high fat and high sodium food intake combined with a sedentary daily routine, driving the worldwide pandemic of overweight (body mass index (BMI) $\geq$ 25 kg/m$^2$) and obesity (BMI $\geq$ 30 kg/m$^2$) [1]. Overweight, obesity and related hypertension are most prevalent in mature adults [2]. Over the last decades, however, especially young adults are gaining weight faster, increasing early-onset hypertension rates (i.e. onset at age below 55 years), and cardiovascular disease risk [3,4].

In young adults, sudden cardiac death (SCD) is predominantly caused by non-ischemic cardiomyopathies [5,6]. When symptoms are non-specific, and initial examinations are inconclusive, cardiac magnetic resonance imaging (MRI) can non-invasively assess volumes, ejection fractions and left ventricular (LV) mass (LVM) to rule-out cardiomyopathy [7]. Important MRI findings include LVM and chamber size alterations, which could be masked by overlapping overweight and hypertension induced cardiac changes, comparable to cardiac remodeling in athletes [8,9]. At early disease stages, this could lead to misdiagnosis and possible SCD. Large cohort studies in adults over 40 years of age demonstrated that increased BMI is associated with increased LVM and cardiac volumes caused by continuous high blood volume circulation, while hypertension is associated with increased LVM caused by elevated systemic vascular resistance [10–12]. In young adults this data is lacking, forcing the use of healthy reference ranges [13]. Indexation of LVM and volumes for body surface area (BSA) is recommended to correct for body size [14]. However, several studies showed that BSA-indexation in overweight individuals masks the presence of left ventricular hypertrophy (LVH), leaving only the effect of hypertension, while other body dimensions could be more appropriate for indexation [15,16].

$T_1$ mapping can determine myocardial changes like fibrosis natively, and extracellular volume (ECV) when repeated after administering contrast-agent [17]. Many cardiomyopathies have an expanded extracellular matrix caused by fibrosis, leading to increased native $T_1$ values and ECV, while myocardial fat infiltration lowers native $T_1$ values [18,19]. In athletes, increased LVM is mainly related to hypertrophy of cardiomyocytes as evidenced by reduced ECV [20]. Detraining in athletes results in a decrease of LVM, comparable to weight loss in overweight and obesity, and antihypertensive treatment in hypertension [21]. $T_2$ mapping measures myocardial water content [17], and these values are often prolonged in cardiomyopathies [22], while these changes are absent in athletes [23], and unknown in overweight and hypertensive young adults.

We hypothesized that in young adults overweight results in increased LVM and volumes, and hypertension in increased LVM. We evaluated the effect of different body scales for indexation. We also hypothesized that the expected increased LVM in young adults is predominantly caused by elevated cellular mass rather than fibrosis, fat or edema.

## Materials and methods

### Study population

This single center study was approved by the local medical ethical committee of the University Medical Center Groningen (no. 2016/476) and complied with the Declaration of Helsinki. All

subjects signed informed consent before participation. We prospectively recruited volunteers aged 18–45 years with at least one cardiac risk factor: overweight (BMI $\geq$ 25 kg/m$^2$) or hypertension, defined as under treatment with antihypertensive medication, or three consecutive blood pressure measurements $\geq$140/90 mmHg. Age- and gender-matched normotensive normal-weight individuals were recruited to serve as controls. Exclusion criteria were type 2 diabetes, cardiac symptoms, history of cardiovascular disease, amateur athletes (physical exercise >3 hours/week) [8], smoking, and standard MRI contraindications.

All included subjects were classified on overweight and hypertension status as: 1. normotensive normal-weight (i.e. controls); 2. normotensive overweight; 3. hypertensive normal-weight; 4. hypertensive overweight. Correct group classification was confirmed with a questionnaire including height, and measurements of weight, blood pressure and HbA1c level. Haematocrit and HbA1c were measured from a blood sample obtained prior to the MRI scan.

## Cardiac MRI

All examinations were performed on a 3T MRI-scanner (MAGNETOM Prisma, Siemens Healthineers, Erlangen, Germany) with a 60-channel phased-array coil. Steady-state free precession (SSFP) sequences were used to acquire multiple long- and short-axis cines covering the entire heart [24]. Mapping data was acquired from a basal, midventricular and apical short-axis slice [17]. $T_1$ mapping was performed using a Modified Look-Locker Inversion Recovery 5(3)3 sequence before and at least 10 minutes after administration of 0.2 mmol/kg Gadoteric acid (Dotarem, Guerbet, France). $T_2$ mapping was performed using a $T_2$-prepared SSFP sequence. Acquisition parameters are provided in S1 Table.

## Image analysis

Image analysis was performed using cvi42 version 5.10.1 (Circle Cardiovascular Imaging, Calgary, Alberta, Canada). Short-axis cines were manually contour-traced for assessment of cardiac morphology and function using a validated protocol [25]. Trabecula were included in the blood pool. As recommended, LVM and volumes were indexed for BSA calculated using the DuBois formula [14], and also for body scales aimed at normalization in overweight, including estimated lean body mass, height, height to the power of 1.7 (height$^{1.7}$) and height$^{2.7}$ [26].

For each short-axis slice, $T_1$ and $T_2$ maps were generated offline using motion-corrected images with different inversion and echo times. All maps were segmented by outlining endocardial and epicardial contours [14]. In $T_1$ maps, a blood pool contour was additionally traced for ECV assessment. Apical segments were excluded from global values, because of artefacts and wide standard deviations in mapping outcomes [27]. The height$^{2.7}$-indexed LVM was used to calculate myocyte mass and extracellular mass [20]:

$$indexed\ myocyte\ mass = height^{2.7} - indexed\ LVM * (100\% - ECV) \qquad (1)$$

$$indexed\ extracellular\ mass = height^{2.7} - indexed\ LVM * ECV \qquad (2)$$

## Statistical analysis

Statistical analysis was performed using SPSS version 24 (IBM SPSS Statistics, Armonk, New York, USA). Continuous variables were reported as mean ± standard deviation. Normality was assessed using the Shapiro-Wilk test and visual inspection of the Q-Q plot [28]. Comparison of two groups was performed using the independent samples t-test, comparison of multiple groups with the analysis of variance and Bonferroni post-hoc testing. Correlations were tested with Pearson correlation. P values < 0.05 were considered significant.

The sample size was calculated using previously published data in athletes [29]. In that study, the mean difference between athletes and controls was ±9 g, and the standard deviation was ±13 g. With a one-sided significance level of 0.05 (i.e., α) and 80% power (i.e., β = 0.20), a sample size of 26 was needed per group.

For some analyses, overweight subjects were further subdivided into mild overweight (BMI 25–29.9 kg/m$^2$) and obese (BMI $\geq$ 30 kg/m$^2$) to further explore possible differences in cardiac MRI parameters.

## Results

One hundred and twenty-six subjects (mean age 35 ± 4 years; 51% males) were included (Table 1). There were no significant differences in age or gender between all four groups.

### Cardiac morphology and function

The normotensive overweight group had significantly higher LVM than controls (P = 0.015), while cardiac and stroke volumes were similar (Table 2 and Fig 1). After subdividing the overweight group into mild overweight and obese, the obese group showed higher LVM (108 ± 24 g) compared to mild overweight (97 ± 19 g, P = 0.483) and controls (87 ± 21 g, P < 0.001) (S2 Table). The obese group also demonstrated non-significantly higher cardiac volumes than controls and mild overweight. The mass-volume ratio in the normotensive overweight group (0.61 ± 0.12 g/ml) was significantly higher compared to controls (0.51 ± 0.09 g/ml) (P < 0.001) (Fig 2). Gender-specific cardiac morphology and function outcomes are reported in S3–S6 Tables.

**Table 1. Study population characteristics.**

|  | Normotensive | | Hypertensive | |
|---|---|---|---|---|
|  | **Normal-weight** | **Overweight** | **Normal-weight** | **Overweight** |
|  | ***n* = 40** | ***n* = 40** | ***n* = 17** | ***n* = 29** |
| Age (years) | 34 ± 4 | 35 ± 4 | 36 ± 3 | 36 ± 4 |
| Gender, male *n* (%) | 20 (50) | 20 (50) | 10 (59) | 14 (48) |
| Height (cm) | 178 ± 8 | 177 ± 9 | 181 ± 10 | 178 ± 10 |
| Weight (kg) | 70 ± 9 | **100 ± 16**\* | 76 ± 11 | **98 ± 12**\* |
| Body mass index (kg/m$^2$) | 22 ± 2 | **32 ± 4**\* | 23 ± 1 | **31 ± 4**\* |
| Body surface area (m$^2$) | 1.9 ± 0.2 | **2.2 ± 0.2**\* | 2.0 ± 0.2 | **2.2 ± 0.2**\* |
| Lean body mass (kg) | 49 ± 9 | **61 ± 13**\* | 53 ± 10 | **60 ± 12**\* |
| Overweight duration (years) | - | 14 ± 9 | - | 11 ± 7 |
| Systolic blood pressure (mmHg) | 117 ± 8 | 122 ± 7 | **141 ± 16**\* | **144 ± 14**\*† |
| Diastolic blood pressure (mmHg) | 79 ± 6 | 81 ± 7 | **95 ± 9**\* | **94 ± 10**\*† |
| Antihypertensive drugs, *n* (%) | 0 (0) | 0 (0) | **10 (59)**\* | **17 (59)**\*† |
| Hypertension duration (years) | - | - | 7 ± 5 | 6 ± 7 |
| HbA1c (mmol/mol) | 33 ± 3 | 34 ± 3 | 32 ± 4 | 34 ± 3 |
| Haematocrit (%) | 40 ± 4 | 42 ± 3 | 43 ± 3 | 42 ± 4 |
| Heart rate (bpm) | 66 ± 12 | 67 ± 12 | 68 ± 8 | **76 ± 12**\*† |

Data reported as mean ± standard deviation. Bonferroni post-hoc tests

\*P < 0.05 vs normotensive normal-weight

†P < 0.05 vs normotensive overweight.

**Table 2. Cardiac morphology, function and tissue characteristics per group.**

| | Normotensive | | Hypertensive | |
|---|---|---|---|---|
| | **Normal-weight** | **Overweight** | **Normal-weight** | **Overweight** |
| **Left ventricle** | | | | |
| Mass (g) | 87 ± 21 | **104 ± 23***  | **109 ± 27***  | **112 ± 26***  |
| Indexed mass (g/m$^{2.7}$) | 18.5 ± 3.6 | **21.9 ± 3.2***  | **21.6 ± 3.0***  | **23.6 ± 4.1***  |
| Myocyte mass (g/m$^{2.7}$) | 14.0 ± 2.9 | **16.9 ± 2.7***  | **16.5 ± 2.3***  | **18.1 ± 3.5***  |
| Extracellular mass (g/m$^{2.7}$) | 4.5 ± 0.8 | **5.1 ± 0.7***  | 5.1 ± 0.9 | **5.5 ± 0.9***  |
| End-diastolic volume (ml) | 171 ± 29 | 173 ± 36 | 171 ± 36 | 168 ± 38 |
| End-systolic volume (ml) | 68 ± 13 | 71 ± 19 | 70 ± 17 | 65 ± 20 |
| Stroke volume (ml) | 103 ± 20 | 103 ± 21 | 100 ± 21 | 102 ± 21 |
| Ejection fraction (%) | 60 ± 4 | 60 ± 5 | 59 ± 4 | 62 ± 5 |
| Mass-volume ratio (g/ml) | 0.51 ± 0.09 | **0.61 ± 0.12***  | **0.64 ± 0.11***  | **0.67 ± 0.09***  |
| **Right ventricle** | | | | |
| End-diastolic volume (ml) | 192 ± 34 | 196 ± 42 | 191 ± 47 | 186 ± 43 |
| End-systolic volume (ml) | 90 ± 18 | 93 ± 25 | 91 ± 30 | 84 ± 25 |
| Stroke volume (ml) | 102 ± 19 | 102 ± 21 | 100 ± 21 | 102 ± 21 |
| Ejection fraction (%) | 53 ± 4 | 53 ± 5 | 53 ± 6 | 55 ± 5 |
| **Global mapping values** | | | | |
| Native T$_1$ (ms) | 1147 ± 30 | 1153 ± 34 | 1151 ± 35 | 1153 ± 37 |
| Extracellular volume (%) | 24.7 ± 2.1 | **23.3 ± 2.1***  | 23.7 ± 1.9 | **23.5 ± 2.5***  |
| T$_2$ (ms) | 39.2 ± 2.1 | 38.5 ± 1.7 | 38.2 ± 1.4 | **37.5 ± 2.2***  |

Data reported as mean ± standard deviation. Bonferroni post-hoc tests

*P < 0.05 vs normotensive normal-weight.

The hypertensive normal-weight group had significantly higher LVM than controls (P = 0.012), while cardiac and stroke volumes were similar (Table 2 and Fig 1). The mass-volume ratio in the hypertensive normal-weight group (0.64 ± 0.11 g/ml) was also significantly higher compared to controls (P < 0.001) (Fig 2).

The hypertensive overweight group had significantly higher LVM than controls (P < 0.001), while cardiac and stroke volumes were similar (Table 2 and Fig 1). The mass-volume ratio in the hypertensive overweight group (0.67 ± 0.09 g/ml) was higher than in the control group (P < 0.001) and normotensive overweight group (P = 0.075) (Fig 2).

Results of cardiac morphology and function in hypertensive populations with normal-weight, mild overweight and obese are reported in S7 Table. Although not significant, in the obese group, LVM was higher and cardiac volumes were lower. After height$^{2.7}$ indexation, the mild overweight and obese groups showed non-significantly higher end-diastolic volumes, stroke volumes and LVM compared to normal-weights.

## Indexation methods

After indexation for BSA, the LVM in the normotensive overweight group was normalized compared to controls, while LVM remained significantly higher in the hypertensive normal-weight group (P = 0.007) (Fig 1 and S8 Table). In the hypertensive overweight group, BSA-indexed LVM remained higher compared to controls, though only significant in the female subgroup (S3 and S4 Tables). In groups with overweight, cardiac and stroke volumes became significantly lower compared to controls after BSA-indexation (all P < 0.01). Indexation with lean body mass showed the same impact as indexation with BSA.

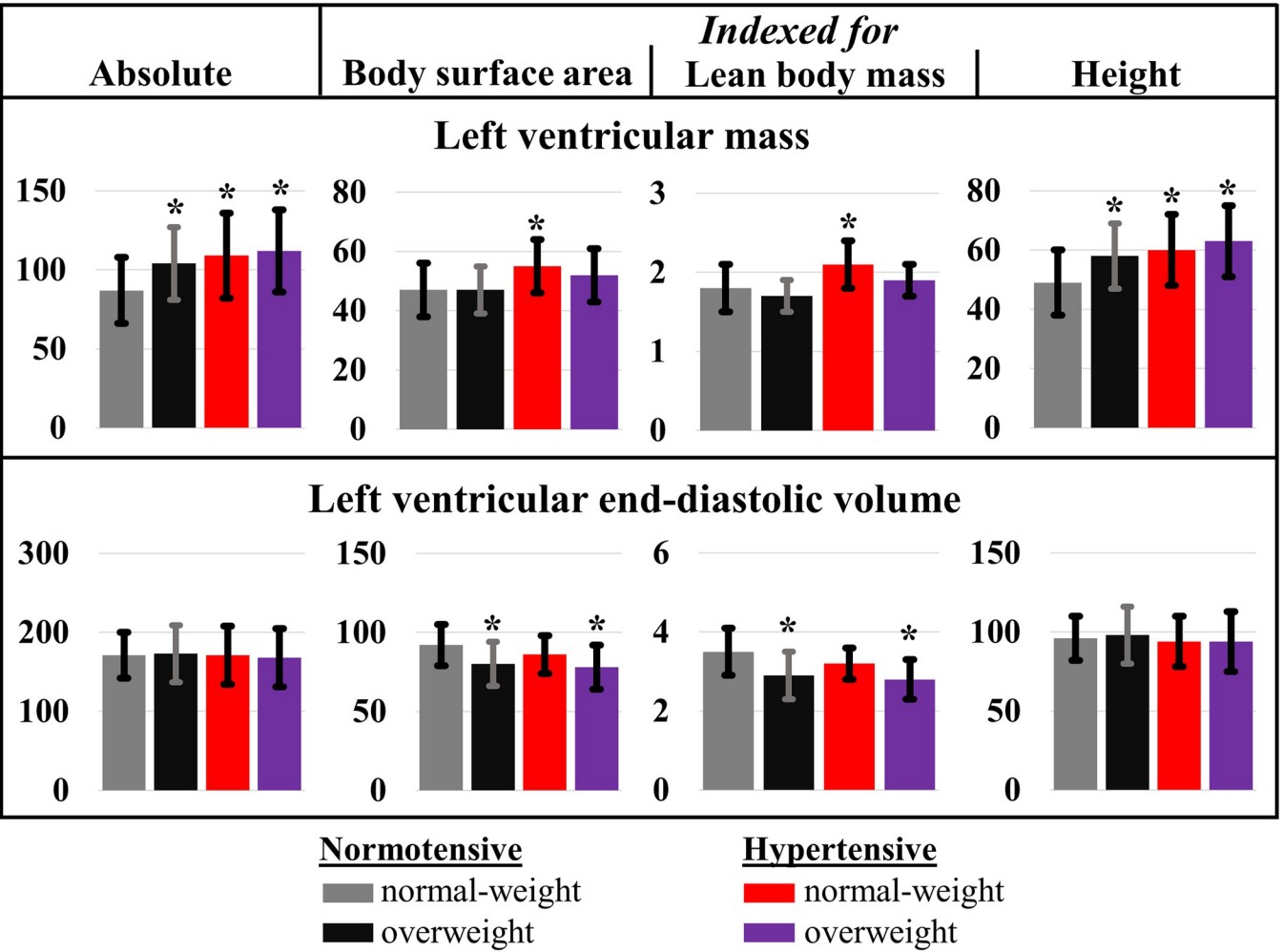

**Fig 1.** Indexation of left ventricular (LV) mass (upper panel) and LV end-diastolic volume (lower panel) for several body scales in the four study groups. Indexation for height[1.7] and height[2.7] were not included in the graph, since these results were similar to the height-indexed results. Bonferroni post-hoc tests: *P < 0.05 vs normotensive normal-weight.

After indexation for height, height[1.7] or height[2.7], LVM remained significantly higher in all risk groups compared to controls (all P < 0.05) (Fig 1 and S8 Table), where cardiac and stroke volumes remained similar.

## Myocardial tissue characteristics

Height[2.7]-indexed LVM and myocardial tissue characteristics are reported in Table 2. Native $T_1$ values were similar between all groups and were not significantly correlated with height[2.7]-indexed LVM, BMI or systolic blood pressure (SBP) (Fig 3 and S1 Fig).

ECV was significantly lower in all risk groups compared to controls (all P < 0.01), except for the normal-weight hypertensive group (P = 0.113). ECV was negatively correlated with height[2.7]-indexed LVM (r = –0.413, P < 0.001), also after adjusting for gender (r = –0.314, P < 0.001) (Fig 3). In normotensive subjects, ECV correlated negatively with BMI (P = 0.020) (S1 Fig). In normal-weight subjects, ECV decreased with increasing SBP (P < 0.001).

$T_2$ values were only significantly lower in the hypertensive overweight group compared to controls (P = 0.004). This difference related to the correlation between $T_2$ values and heart rate

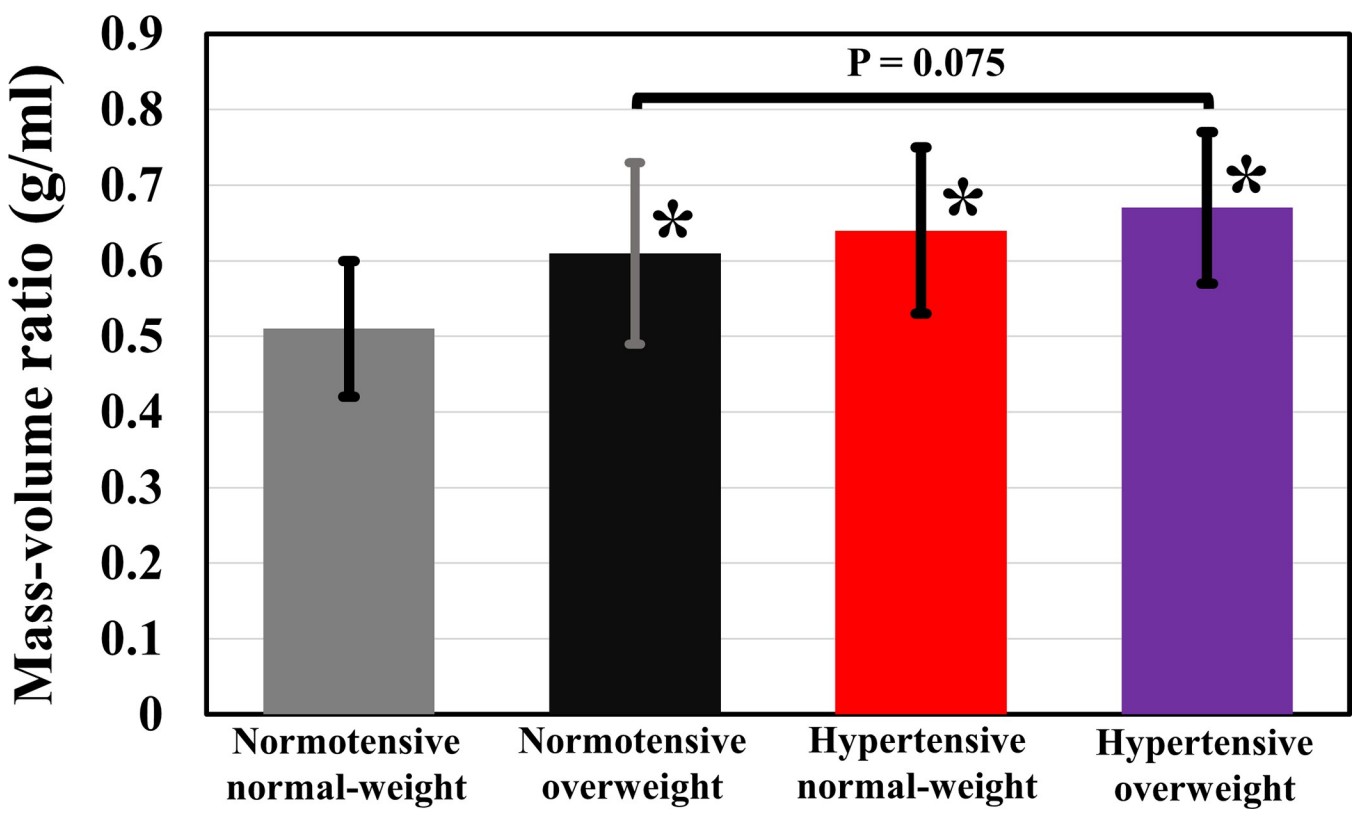

**Fig 2. The mass-volume ratio per group visualized as mean with one standard deviation.** Mass-volume ratio was defined as the ratio between the left ventricular (LV) mass and LV end-diastolic volume. Bonferroni post-hoc tests: *P < 0.001 vs normotensive normal-weight controls.

(r = –0.57, P < 0.001), as only the hypertensive overweight group showed significantly higher heart rate than controls (P = 0.002). No correlation was found between $T_2$ values and height$^{2.7}$-indexed LVM (Fig 3).

In the normotensive overweight group, the height$^{2.7}$-indexed LVM was 19% higher compared to controls (P < 0.001) (Fig 4). This increase was predominantly caused by elevated myocyte mass relative to controls (+20%, P < 0.001), and to a lesser extent by extracellular mass growth (+12%, P = 0.024). In the hypertensive normal-weight group, the height$^{2.7}$-

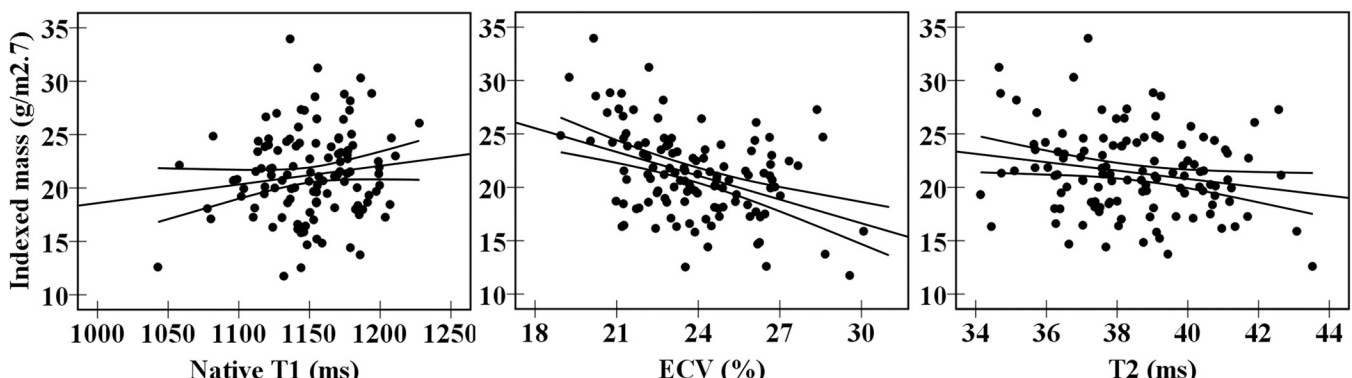

**Fig 3. Correlation between myocardial tissue characteristics and left ventricular mass indexed for height to the power of 2.7.** From left to right, native $T_1$ mapping (r = 0.152, P = 0.093), extracellular volume (ECV) (r = –0.413, P < 0.001) and $T_2$ mapping (r = –0.168, P = 0.063).

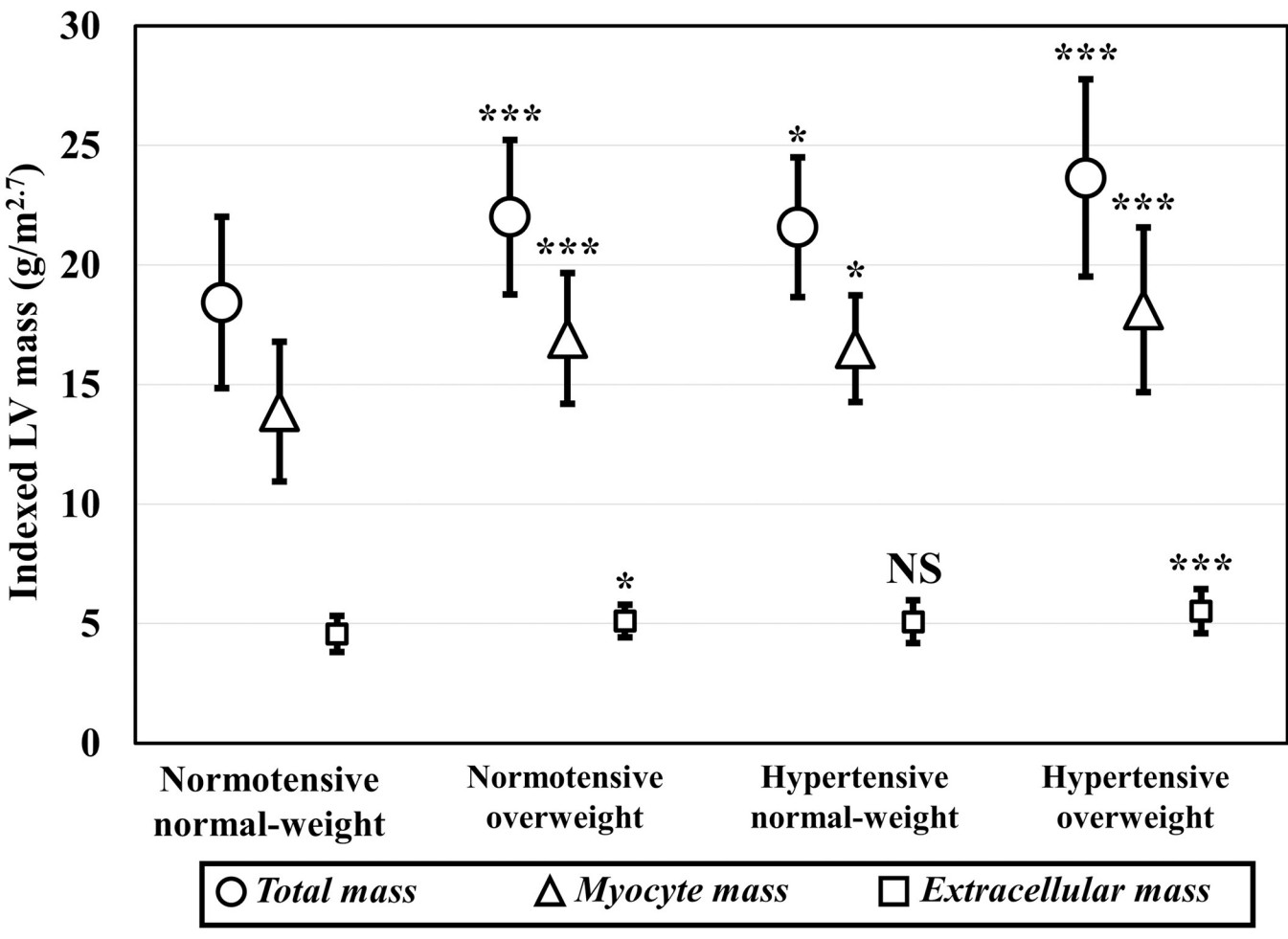

**Fig 4. Left ventricular (LV) mass indexed for height to the power of 2.7 visualized as mean with standard deviation.** The different markers represent total myocardial mass, myocyte mass and extracellular mass. Significantly higher values compared to the normotensive normal-weight group are $P < 0.05$ (*), $P < 0.01$ (**) and $P < 0.001$ (***). *NS* not significant.

indexed LVM was 17% higher than in controls ($P = 0.017$), and related to an increase of 18% in myocyte mass ($P = 0.014$) and 13% in extracellular mass ($P = 0.185$). In the hypertensive overweight group, the height$^{2.7}$-indexed LVM was 28% higher compared to controls ($P < 0.001$), accompanied by an increase of 30% myocyte mass and 22% extracellular mass (both $P < 0.001$).

## Discussion

This study shows that young adults with overweight and hypertension have increased LVM, predominantly caused by hypertrophy of cardiomyocytes instead of fibrosis or fatty infiltration, as evidenced by similar $T_1$ values and lower ECV. Indexation using the weight-related body scales BSA and lean body mass normalized LVM and overly lowered cardiac volumes in overweight groups compared to controls, while LVM remained higher in the hypertensive normal-weight group.

In young adults with overweight and/or hypertension, the increased LVM was highest when both conditions were present. Our findings in overweight groups were consistent with previous studies reporting increasing LVM with higher BMI versus controls; in mature adults

(aged 61 ± 9 years, no sex-specific results) [10–12,30], in young adult obese males (aged 30 ± 7 years, BMI 35 ± 3 kg/m$^2$) [31], and in severely obese adolescents (aged 18 ± 4 years, BMI 41 ± 6 kg/m$^2$) [32]. Increased LVM in hypertensives is thoroughly reported, however, available studies included only mature adults (mean age, range 49–61 years) as hypertension at younger age is less prevalent [33–36]. Some of these studies classified hypertensives into a group with and without LVH to investigate differences in underlying myocardial tissue properties [33,34]. The combined effect of overweight and hypertension in young adults as observed in the current study is in line with findings in mature populations (age range 45–84 years) [10,12].

Overweight subjects have higher circulating blood volume than normal-weights to meet increased metabolic demands [37]. We found correspondingly higher cardiac volumes in our obese groups, however, not in all mild overweight groups. Previous studies confirm our findings in the obese for different age groups, including adolescents [31,38,39]. Another study showed increasing cardiac and stroke volumes with increasing BMI categories, however, comparable to our findings, these increases were not significant in the group with mild overweight [30].

Long-term increased mass-volume ratio is associated with major adverse cardiovascular events in asymptomatic populations [40]. Diet aimed at weight loss and antihypertensive treatment is often initiated, resulting in decreased concentric LV remodeling [21]. Confirming literature, we found a higher mass-volume ratio in our young mild overweight, obese and hypertensive subjects [30,31,33,34,39].

The increased absolute LVM in overweight subjects was normalized after BSA-indexation, causing an undesirable underestimation of LVH which is also in line with previous studies [15,41]. With indexation for any height variable, the absolute LVM remained higher in overweight subjects. In a previous study, height$^{2.7}$ was stated as best normalization method for identification of LVH as it was most closely associated with adverse outcomes [42]. Moreover, in the echocardiographic guideline, height$^{2.7}$-indexation was already acknowledged as standard, while BSA-indexation remained acceptable in normal-weight patients [43]. We also showed that volumes were underestimated in overweight groups after BSA-indexation, confirmed by one other study [15]. Considering these findings, additionally reporting height$^{2.7}$-indexed values in clinical reports seems advisable. Furthermore, as BSA-indexation normalizes the increased LVM in overweight populations without cardiovascular disease, this could potentially help differentiating between LVM adaptation to overweight and cardiomyopathy.

Our results suggest that increased LVM in young adults with overweight and/or hypertension is mainly caused by increased myocyte mass and to a lesser extent by fibrosis, as evidenced by similar $T_1$ values and lower ECV. This is confirmed by a study in athletes that reported 30% higher myocyte mass and 16% higher extracellular mass relative to controls [20]. Two studies have reported on $T_1$ mapping and ECV in obese young adults [39,44]. One study in obese young adults (aged 31 ± 6 years, BMI 33 ± 2 kg/m$^2$) showed $T_1$ values, ECV, and LVM comparable to controls [44]. In the other study, severely obese adolescents (aged 18 ± 4 years, BMI 41 ± 6 kg/m$^2$) showed increased ECV and LVM, contradicting our findings, possibly related to the greater overweight severity of their study population [39].

Previous studies in more mature adults (mean age, range 49–61 years) with hypertension and LVH showed increased native $T_1$ values and ECV, suggesting a component of interstitial fibrosis, probably related to the duration of this condition [33,34,45], as without LVH, $T_1$ values and ECV were mostly similar to controls.

To our best knowledge, this study is the first to report $T_2$ mapping values in overweight and hypertensive populations [22]. Only the hypertensive overweight group showed lower $T_2$ values than controls, corresponding to increased heart rates in this group, a known effect in SSFP-based $T_2$ mapping [46].

This study has several limitations. First, we included all hypertensive subjects in one group, irrespective of anti-hypertensive treatment. However, before hypertensives with and without treatment were merged, all cardiac outcomes showed no significant differences (S9 Table). Second, only asymptomatic subjects without known cardiovascular disease were included. Results could therefore not be compared directly to patients with known cardiac disease. Third, this is a cross-sectional study without follow-up, therefore long-term consequences of the demonstrated cardiac alterations could not be investigated. Fourth, the hypertensive normal-weight group was smaller than the sample size calculated with the power analysis.

## Conclusion

Overweight and hypertension increase LVM in young adults, mainly by hypertrophy of cardiomyocytes instead of fibrosis or fatty infiltration. In overweight, BSA indexation results in normalization of LVM and smaller cardiac volumes than controls. Indexation with height[2.7] circumvents this effect and seems advisable.

## Supporting information

**S1 Fig. Correlations between risk factors and tissue characteristics.** In the upper panel, only normotensive subjects are included to show the correlation between body mass index (BMI) and native $T_1$ (P = 0.170), and between BMI and extracellular volume (ECV) (r = –0.271, P = 0.020). In the lower panel, only normal-weight subjects are included to show the correlation between systolic blood pressure (SBP) and native $T_1$ (P = 0.751), and between SBP and ECV (r = –0.471, P < 0.001).
(TIF)

**S1 Table. Acquisition parameters.**
(DOCX)

**S2 Table. Cardiac morphology and function in normotensive subjects divided on BMI.**
Data reported as mean ± standard deviation. *P < 0.05 versus normal-weight, †P < 0.05 versus mild overweight *BMI* body mass index.
(DOCX)

**S3 Table. Cardiac morphology and function per male subgroup.** Data reported as mean ± standard deviation (SD) (range for reference). Ranges for reference were calculated as mean ± $t_{0.975,n-1}$ · $\sqrt{((n+1)/n)}$ · SD. *P < 0.05 versus normotensive normal-weight *EDV* end-diastolic volume, *ESV* end-systolic volume, *SV* stroke volume, *EF* ejection fraction, *BSA* body surface area.
(DOCX)

**S4 Table. Cardiac morphology and function per female subgroup.** Data reported as mean ± standard deviation (SD) (range for reference). Ranges for reference were calculated as mean ± $t_{0.975,n-1}$ · $\sqrt{((n+1)/n)}$ · SD. *P < 0.05 versus normotensive normal-weight, †P < 0.05 versus normotensive overweight *EDV* end-diastolic volume, *ESV* end-systolic volume, *SV* stroke volume, *EF* ejection fraction, *BSA* body surface area.
(DOCX)

**S5 Table. Cardiac morphology and function in normotensive males divided on BMI.** Data reported as mean ± standard deviation. *P < 0.05 versus normal-weight, †P < 0.05 versus mild overweight *BMI* body mass index.
(DOCX)

**S6 Table. Cardiac morphology and function in normotensive females divided on BMI.**
Data reported as mean ± standard deviation. *P < 0.05 versus normal-weight, †P < 0.05 versus mild overweight *BMI* body mass index.
(DOCX)

**S7 Table. Cardiac morphology and function in hypertensive subjects divided on BMI.** Data reported as mean ± standard deviation. *P < 0.05 versus normal-weight, †P < 0.05 versus mild overweight *BMI* body mass index.
(DOCX)

**S8 Table. Indexation of volumes and mass for different body scales.** Data reported as mean ± standard deviation. *P < 0.05 versus normotensive normal-weight.
(DOCX)

**S9 Table. Differences between hypertensive subjects on presence anti-hypertensive medication.** Data reported as mean ± standard deviation. *Indexed* indexed with body surface area, *EDV* end-diastolic volume, *ESV* end-systolic volume, *SV* stroke volume.
(DOCX)

## Author Contributions

**Conceptualization:** Riemer H. J. A. Slart, Birgitta K. Velthuis, Christopher T. Nguyen, David E. Sosnovik, Vincent M. van Deursen, Niek H. J. Prakken.

**Data curation:** Gert J. H. Snel, Niek H. J. Prakken.

**Formal analysis:** Gert J. H. Snel.

**Funding acquisition:** Rudi A. J. O. Dierckx, Niek H. J. Prakken.

**Investigation:** Maaike van den Boomen, Niek H. J. Prakken.

**Project administration:** Gert J. H. Snel, Niek H. J. Prakken.

**Resources:** Rudi A. J. O. Dierckx, Ronald J. H. Borra.

**Software:** Maaike van den Boomen, Ronald J. H. Borra.

**Supervision:** Riemer H. J. A. Slart, Ronald J. H. Borra, Niek H. J. Prakken.

**Writing – original draft:** Gert J. H. Snel, Niek H. J. Prakken.

**Writing – review & editing:** Riemer H. J. A. Slart, Birgitta K. Velthuis, Maaike van den Boomen, Christopher T. Nguyen, David E. Sosnovik, Vincent M. van Deursen, Rudi A. J. O. Dierckx, Ronald J. H. Borra.

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
