## [Decision Letter · Decision Letter 0]

21 Jun 2022

PONE-D-21-38012Interpretation of pre-morbid cardiac 3T MRI findings in overweight and hypertensive young adults.PLOS ONE

Dear Dr. Snel,

Thank you for submitting your manuscript to PLOS ONE. After careful consideration, we feel that it has merit but does not fully meet PLOS ONE’s publication criteria as it currently stands. Therefore, we invite you to submit a revised version of the manuscript that addresses the points raised during the review process.

We look forward to receiving your revised manuscript.

Kind regards,

Joshua M. Hare, M.D., F.A.C.C., F.A.H.A.

Academic Editor

PLOS ONE

Journal Requirements:

When submitting your revision, we need you to address these additional requirement.

Additional Editor Comments (if provided):

The manuscript by Snel et al. discusses a novel approach toward measuring cardiac structure in overweight individuals. Additional analysis of these normalization methods regarding LV volumes are needed. There was also some concern regarding the classification of some overweight individuals as obese in only some analyses. A discussion of the difference of the analysis between males and females is also needed.

Reviewers' comments:

Reviewer's Responses to Questions

**Comments to the Author**

1. Is the manuscript technically sound, and do the data support the conclusions?

Reviewer #1: Yes

2. Has the statistical analysis been performed appropriately and rigorously? 

Reviewer #1: Yes

3. Have the authors made all data underlying the findings in their manuscript fully available?

Reviewer #1: Yes

4. Is the manuscript presented in an intelligible fashion and written in standard English?

Reviewer #1: Yes

5. Review Comments to the Author

Reviewer #1: Snel et al. used MRI analysis to examine cardiac structure in normal and overweight normotensive and hypertensive young adults. They conclude that increased LV mass is masked when using BSA as the normalizing factor and that using height2.7 does not. The increase in the number of overweight and obese young people make this topic particularly relevant. There are some comments to be addressed.

Major comments:

1. Obese participants are only reported separately for some comparisons but are combined with the overweight group in most comparisons. What is the rationale for this limited analysis and can the authors differentiate between obese and overweight participants in additional studies? The majority of overweight normotensives are obese, what was the number of obese vs. overweight hypertensive.

2. The authors discuss results from obese males. Are these results reported in a Table? Please indicate the number of obese females is not indicated. the number of obese males and females.

3. Table S4 shows that BSA masks LV mass but accentuates differences in volumes (ESV, EDV, SV), particularly in obese patients and to a lesser extent RV volumes (EDV; SV in obese). In Table S6 male normotensives overall follow this pattern in overweight normotensives but Table S8 shows that in females, BSA masks mass in normotensive but not in hypertensives and volumes are accentuated in both overweight groups. Please elaborate

4. 4. In Table 2, differences in the global mapping the values extracellular volume and T2 are virtually identical in overweight normotensive and both hypertensive groups, although, some are significant. What is the relevance of these differences?

Minor comments:

1. Please define the parameters for obese in the Methods section

2. In Table S8 it states that “data reported as mean ± SD”. Please similarly indicate, either in the Methods section or in the legend for each table.

6. PLOS authors have the option to publish the peer review history of their article (what does this mean?). If published, this will include your full peer review and any attached files.

Reviewer #1: No

---

## [Author Response · Author response to Decision Letter 0]

11 Jul 2022

Dear editor and reviewer,

Thank you for handling our manuscript and for providing helpful comments.

Please find below our point-to-point response to each comment.

As requested, we uploaded a marked-up copy of the manuscript with changes tracked and an unmarked version without changes tracked.

Also, we uploaded the newest versions of the supplementary materials.

Your sincerely,

Gert Jan Snel – Corresponding author

Major comments:

1. Obese participants are only reported separately for some comparisons but are combined with the overweight group in most comparisons. What is the rationale for this limited analysis and can the authors differentiate between obese and overweight participants in additional studies? The majority of overweight normotensives are obese, what was the number of obese vs. overweight hypertensive.

We thank the reviewer for these questions. One of our hypotheses was that being overweight without hypertension would result in measurably larger cardiac volumes compared to being normal weight. The cardiac adaptation in our normotensive overweight population was non-significantly different from controls. We therefore performed an analysis in which we subdivided our normotensive overweight group into mild overweight and obese. 

We agree that it would also be of interest to subdivide the hypertensive overweight population (n = 29) into mild overweight (n = 14) and obese (n = 15). We compared these new subgroups to our hypertensive normal-weight group and investigated the impact of overweight in these subjects. The differences between these groups were all non-significant and are added in supplementary tables (S14 and S15). The following text is added to the section Results – Cardiac morphology and function:

“Results of cardiac morphology and function in hypertensive populations with normal-weight, mild overweight and obese are reported in S14 and S15 Tables. Although not significant, in the obese group, LVM was higher and cardiac volumes were lower. After height2.7 indexation, the mild overweight and obese groups showed non-significantly higher end-diastolic volumes, stroke volumes and LVM compared to normal-weights.”

2. The authors discuss results from obese males. Are these results reported in a Table? Please indicate the number of obese females is not indicated. the number of obese males and females.

Thank you for this clarification request. Results from obese males were indeed not reported in a table. We added supplementary tables to report the results of both obese males (n = 13, Tables S10 and S11) and obese females (n = 12, Tables S12 and S13). We referred to these tables in the section Results – Cardiac morphology and function.

For clarity, we rephrased the sentence of the discussion to state:

“Our findings in overweight groups were consistent with previous studies reporting increasing LVM with higher BMI versus controls; in mature adults (aged 61 ± 9 years, no sex-specific results) [10–12,29], in young adult obese males (aged 30 ± 7 years, BMI 35 ± 3 kg/m2) [30], and in severely obese adolescents (aged 18 ± 4 years, BMI 41 ± 6 kg/m2) [31].”

3. Table S4 shows that BSA masks LV mass but accentuates differences in volumes (ESV, EDV, SV), particularly in obese patients and to a lesser extent RV volumes (EDV; SV in obese). In Table S6 male normotensives overall follow this pattern in overweight normotensives but Table S8 shows that in females, BSA masks mass in normotensive but not in hypertensives and volumes are accentuated in both overweight groups. Please elaborate

In both normotensive overweight and hypertensive overweight groups, BSA-indexation indeed resulted in lower volumes compared to normal-weights. This pattern was observed in the entire population (Table S4 and S14), and separately for males (Table S6) and females (Table S8).

BSA-indexation masked LV mass in normotensive overweights as well. There was less masking of LV mass in hypertensive overweights because of the additional cardiac adaptation to hypertension. As a result, BSA-indexed LV mass remained higher in both hypertensive overweight males (Table S6) and females (Table S8) compared to controls, though only significant in females. 

We added the following clarifying sentence in Results – Indexation Methods:

“In the hypertensive overweight group, BSA-indexed LVM remained higher compared to controls, though only significant in the female subgroup (S6 and S8 Tables).”

We also rephrased one sentence in the abstract to state:

“Body surface area (BSA) indexation resulted in LVM normalization in overweights (48 ± 8 g/m2) versus controls (47 ± 9 g/m2), but not in hypertensives (55 ± 9 g/m2) and hypertensive overweights (52 ± 9 g/m2).”

4. 4. In Table 2, differences in the global mapping the values extracellular volume and T2 are virtually identical in overweight normotensive and both hypertensive groups, although, some are significant. What is the relevance of these differences?

Thank you for pointing this out. We originally rounded T2 and extracellular volume (ECV) up to the closest integer, which made these results appear more identical and statistically relevant differences less apparent. We added one decimal for T2 and ECV in Tables 2 and S18.

T2 mapping values were only significantly different in the hypertensive overweight group compared to controls, because of higher heart rates causing lower T2 values, which is a known effect in SSFP-based sequences, as stated in the discussion (R256-258).

All risk groups showed lower ECV values compared to controls, though not significant in the normal-weight hypertensive group. These changes suggest that the increased LV mass mainly related to hypertrophy of myocytes, and to a lesser extent fibrosis (since T1 values were similar). We rephrased one sentence in the discussion to state:

“Our results suggest that increased LVM in young adults with overweight and/or hypertension is mainly caused by increased myocyte mass and to a lesser extent by fibrosis, as evidenced by similar T1 values and lower ECV.”

Minor comments:

1. Please define the parameters for obese in the Methods section

Thank you for this request. In the inclusion criteria of the study, we used a BMI of 25 kg/m2 as cut-off to classify subjects as either normal-weight or overweight. Since we considered the severity of overweight to be potentially of interest during the analysis, we subdivided the overweight population into mild overweight and obese to explore possible differences. Therefore, we defined mild overweight and obese in the section Methods – Statistical Analysis by adding the following text: 

“For some analyses, overweight subjects were further subdivided into mild overweight (BMI 25–29.9 kg/m2) and obese (BMI ≥ 30 kg/m2) to further explore possible differences in cardiac MRI parameters.”

In the section Results – Cardiac morphology and function, we removed “BMI 25-29.9 kg/m2” and “BMI 30-43 kg/m2”, to avoid redundancy after defining these groups in the Methods section. Furthermore, we renamed overweight (BMI 25–29.9 kg/m2) as mild overweight to prevent confusion with the original overweight group as stated in inclusion criteria (BMI ≥ 25 kg/m2).

2. In Table S8 it states that “data reported as mean ± SD”. Please similarly indicate, either in the Methods section or in the legend for each table.

In the original manuscript, we stated in the section Methods – Statistical analysis that “Continuous variables were reported as mean ± standard deviation”. We agree that it would be clearer to also include this in the table legends. We added this information in all relevant tables; 1, 2, S4, S16 and S18.

---

## [Decision Letter · Decision Letter 1]

24 Oct 2022

PONE-D-21-38012R1Interpretation of pre-morbid cardiac 3T MRI findings in overweight and hypertensive young adults.PLOS ONE

Dear Dr. Snel,

Thank you for submitting your manuscript to PLOS ONE. After careful consideration, we feel that it has merit but does not fully meet PLOS ONE’s publication criteria as it currently stands. Therefore, we invite you to submit a revised version of the manuscript that addresses the points raised during the review process.

We look forward to receiving your revised manuscript.

Kind regards,

Eduard Shantsila

Academic Editor

PLOS ONE

Journal Requirements:

Additional Editor Comments (if provided):

This is an interesting and well-conducted study. The comments of the original reviewer have been adequately addressed. The comments of the new reviewer are relatively minor and should be easily addressable.

Reviewers' comments:

Reviewer's Responses to Questions

**Comments to the Author**

1. If the authors have adequately addressed your comments raised in a previous round of review and you feel that this manuscript is now acceptable for publication, you may indicate that here to bypass the “Comments to the Author” section, enter your conflict of interest statement in the “Confidential to Editor” section, and submit your "Accept" recommendation.

Reviewer #1: All comments have been addressed

Reviewer #2: (No Response)

2. Is the manuscript technically sound, and do the data support the conclusions?

Reviewer #1: Yes

Reviewer #2: Yes

3. Has the statistical analysis been performed appropriately and rigorously? 

Reviewer #1: Yes

Reviewer #2: Yes

4. Have the authors made all data underlying the findings in their manuscript fully available?

Reviewer #1: Yes

Reviewer #2: Yes

5. Is the manuscript presented in an intelligible fashion and written in standard English?

Reviewer #1: Yes

Reviewer #2: Yes

6. Review Comments to the Author

Reviewer #1: (No Response)

Reviewer #2: Title

Interpretation of pre-morbid cardiac 3T MRI findings in overweight and hypertensive

young adults.

Synopsys

This work aimed to assess whether overweight and hypertension could induce increased left ventricular mass (LVM) and myocardial hypertrophy in young adults. The authors hence conducted a prospective study enrolling 126 subjects (40 age- and sex-matched healty controls, 40 normotensive overweight, 17 hypertensive and 29 hypertensive overweight). It was observed that LVM increased in all risk groups compared to controls, and that indexation by body surface area (BSA) led to LVM normalization in overweights but not in the hypertensive groups. Furthermore, BSA indexation underestimated ventricular volumes in overweight patients, whereas height^2.7 did not. Finally, the authors observed a lower myocardial extracellular volume (ECV) in all risk groups, whereas myocardial native and post-contrast T1 and T2 did not show significative alterations, suggesting myocytes hypertrophy.

Strengths

• Prospective design

• Very detailed report on all the analysis performed

Weaknesses

• No sample size calculation reported

• High number of supplementary tables results in a dispersive fruition

Abstract

1. Please state explicitly the aims of your work in the abstract within the first sentences.

Introduction

2. Page 3 lines 63-64: perhaps the word “volume” is missing in “cardiac volumes caused by continuous high blood circulation”.

Materials and methods

3. Please report a sample size calculation if such analysis was performed in the study planning stages. Otherwise, I would suggest conducting a statistical power analysis, given the high number of comparisons performed and the relatively small sizes of the study groups.

Results

4. While I appreciate the detailed report on all the analysis performed, I found very dispersive to scroll between 18 different supplementary tables. Therefore, I would suggest the authors to reduce such number, perhaps by collapsing similar tables into one or by avoiding to report non-fundamental data.

7. PLOS authors have the option to publish the peer review history of their article (what does this mean?). If published, this will include your full peer review and any attached files.

Reviewer #1: No

Reviewer #2: No

---

## [Author Response · Author response to Decision Letter 1]

4 Nov 2022

Dear editor and reviewers,

Thank you for handling our manuscript and providing helpful comments.

Please find below our point-to-point response to each comment.

As requested, we uploaded a marked-up copy of the manuscript with tracked changes and an unmarked version.

Also, we uploaded the newest versions of the supplementary materials.

Your sincerely,

Gert Jan Snel – Corresponding author

Reviewer #1: (No Response)

Reviewer #2: 

Abstract

1. Please state explicitly the aims of your work in the abstract within the first sentences.

Thank you for this clarification request. We rephrased the first sentences in the abstract to state:

“In young adults, overweight and hypertension possibly already trigger cardiac remodeling as seen in mature adults, potentially overlapping non-ischemic cardiomyopathy findings. To this end, in young overweight and hypertensive adults, we aimed to investigate changes in left ventricular mass (LVM) and cardiac volumes, and the impact of different body scales for indexation. We also aimed to explore the presence of myocardial fibrosis, fat and edema, and changes in cellular mass with extracellular volume (ECV), T1 and T2 tissue characteristics.”

Introduction

2. Page 3 lines 63-64: perhaps the word “volume” is missing in “cardiac volumes caused by continuous high blood circulation”.

Thank you for pointing this out. We added the word “volume” in that sentence to state: 

“Large cohort studies in adults over 40 years of age demonstrated that increased BMI is associated with increased LVM and cardiac volumes caused by continuous high blood volume circulation, while hypertension is associated with increased LVM caused by elevated systemic vascular resistance [10–12].”

Materials and methods

3. Please report a sample size calculation if such analysis was performed in the study planning stages. Otherwise, I would suggest conducting a statistical power analysis, given the high number of comparisons performed and the relatively small sizes of the study groups.

Thank you for this request. We indeed performed a sample size analysis in the study planning stages. We added the following paragraph to the “Statistical Analysis” section:

“The sample size was calculated using previously published data in athletes [29]. In that study, the mean difference between athletes and controls was ±9 g, and the standard deviation was ±13 g. With a one-sided significance level of 0.05 (i.e., α) and 80% power (i.e., β=0.20), a sample size of 26 was needed per group.”

Our group sizes of controls, normotensive overweight and hypertensive overweight were sufficiently large. Due to the relatively low prevalence of hypertension in the general normal-weight population between 18 and 45 years of age, our included hypertensive normal-weight group size was smaller than the number calculated in our power analysis. We added the smaller group size as a limitation by stating:

“Fourth, the hypertensive normal-weight group was smaller than the sample size calculated with the power analysis.” 

Results

4. While I appreciate the detailed report on all the analysis performed, I found very dispersive to scroll between 18 different supplementary tables. Therefore, I would suggest the authors to reduce such number, perhaps by collapsing similar tables into one or by avoiding to report non-fundamental data.

We thank the reviewer for this suggestion. We removed the supplementary tables containing solely P-values since the fundamental data and significant differences are reported in other tables. As a result, the number of supplementary tables was reduced from 18 to 9.

---

## [Editor Report · Decision Letter 2]

15 Nov 2022

Interpretation of pre-morbid cardiac 3T MRI findings in overweight and hypertensive young adults.

PONE-D-21-38012R2

Dear Dr. Snel,

We’re pleased to inform you that your manuscript has been judged scientifically suitable for publication and will be formally accepted for publication once it meets all outstanding technical requirements.

Kind regards,

Eduard Shantsila

Academic Editor

PLOS ONE

---

## [Editor Report · Acceptance letter]

22 Nov 2022

PONE-D-21-38012R2 

Interpretation of pre-morbid cardiac 3T MRI findings in overweight and hypertensive young adults. 

Dear Dr. Snel:

I'm pleased to inform you that your manuscript has been deemed suitable for publication in PLOS ONE. Congratulations! Your manuscript is now with our production department. 

Kind regards, 

on behalf of

Dr. Eduard Shantsila 

Academic Editor

PLOS ONE